# A multi-factor dynamic time series measure for stock correlation analysis

**Jinyu Fan[1,2], Guanyu Lu[3], Jun Ma[1,2]***

**1** Qinghai Normal University, Xining, China, **2** Qinghai University, Xining, China, **3** Beijing Jiaotong University, Beijing, China

\* mjun7302@qhu.edu.cn

## Abstract

Existing similarity measures in stock correlation analysis often overlook the multi-dimensional nature of stock data and the dynamics of the time-lag effect (TLE) in phase differences. To address these limitations, this paper proposes a novel method, called Multi-Factor Dynamic Temporal Similarity Measure (MFDTSM). The method introduces an enhanced eXtreme Gradient Boosting (XGBoost) model based on Shapley Additive exPlanations (SHAP) to comprehensively evaluate the influence of stock factors. The proposed method effectively categorizes stocks and reveals the heterogeneity of factor influence by clustering the SHAP values of stock factors. Furthermore, MFDTSM is able to successfully quantify the dynamic rate of phase differences in TLE by constructing the cumulative distance matrix and analyzing the optimal alignment paths of time series data, thereby significantly improving the accuracy of the similarity measure. Empirical analysis is performed using 102 stocks from the communication and financial industries, including 12 key stock factors. The experimental results demonstrate that MFDTSM improves the accuracy of the analysis of industry correlation, linear correlation, and stock correlation pricing by 10%, 16%, and 5%, respectively, over existing methods, which highlight the efficiency and stability of MFDTSM in analyzing complex stock market dynamics.

## Introduction

Analyzing stock correlation is essential for the study of financial market [1]. This analysis enhances investors' and analysts' understanding of market dynamics while providing a solid scientific foundation for precise investment strategies [2]. Through in-depth analysis of stock correlations, it can effectively predict overall market trends and individual stock price fluctuations, thereby assisting in building a lower-risk, more multidimensional portfolio, as well as exploring potential value investment opportunities [3]. Therefore, optimizing the accuracy and performance of stock similarity measures directly improves market participants' understanding of stock market dynamics and predicting ability [4].

**Data availability statement:** The data underlying this study were sourced from JoinQuant under a commercial license for the JQData service. Due to licensing restrictions, the authors are not permitted to redistribute the raw data. These data are available for purchase directly from JoinQuant. Interested researchers can contact the official JoinQuant operations team to inquire about and procure the JQData service packages via their official business email (jqdatasdk@joinquant.com).

**Funding:** The author(s) received no specific funding for this work.

**Competing interests:** The authors have declared that no competing interests exist.

Stock market data analysis reveals two key features: *data multidimensionality* [5] and the *dynamics of the Time-lag Effect (TLE)* [6]. Firstly, the multidimensionality of stock market data contains a variety of factors including price, turnover rate, and return on assets [7], which reflect the complexity of market behavior from different perspectives [8,9]. However, most existing stock similarity measures, such as Euclidean distance (ED) [10] and Pearson product-moment correlation coefficient (PPC) [11], focus solely on the analysis of single-factor data and are unable to fully exploit these multidimensional data for comprehensive correlation assessment. Thus, they neglect the joint impact of different factors on stock price volatility. Secondly, TLE indicates the asynchrony in stock price fluctuations, which is characterized by phase differences in the time domain [12]. For example, during a global chip shortage, semiconductor companies tend to lead electronics manufacturers in stock price, and this lead changes dynamically over time, suggesting that the phase difference has a dynamic nature [13]. Although methods such as Longest Common Subsequence (LCSS) [14] and Dynamic Time Warping (DTW) [15] can manage scaling and misalignment in time series, they are limited in capturing the dynamic changes in phase differences within TLE. In recent years, deep learning approaches, particularly neural network-based models such as Long Short-Term Memory (LSTM) , Gated Recurrent Units (GRU) , and Transformers , have also been widely adopted for time series analysis due to their ability to capture complex temporal dependencies and nonlinear patterns. For instance, LSTMs have been successfully applied in stock price prediction and portfolio optimization, while attention mechanisms enhance the model's focus on salient time steps. However, these methods often require large amounts of data and computational resources, and their black-box nature can limit interpretability in factor-specific analysis. Although neural networks excel in pattern recognition, they are less focused on explicitly modeling factor influence and dynamic phase shifts in time-lag effects, which are central to our proposed MFDTSM framework.

To address these challenges, we propose a novel measure called the *Multi-Factor Dynamic Temporal Similarity Measure (MFDTSM)*. Specifically, MFDTSM employs an XGBoost (eXtreme Gradient Boosting) [16] model augmented with SHAP (SHapley Additive exPlanations) [17] to analyze the complex non-linear relationships among multidimensional factors and their impacts on stock market dynamics. SHAP values are calculated to quantify the importance of the factor and clustering based on SHAP is performed to group stocks with similar patterns of factor influence, reducing computational redundancy by eliminating unnecessary calculations on dissimilar stocks.

Additionally, we construct a cumulative distance matrix to explore phase differences and their dynamic evolution in TLE, thus optimizing similarity measures between stock time series. In the first step, the matrix determines the optimal alignment path of the stock time series [18], and then quantifies the rate of phase difference variation by analyzing the direction change of the coordinate points on the path [19]. This approach enables accurate capture of the best match between time series, while providing an advanced methodology for revealing the dynamics of time series phase differences. In this paper, we develop the innovative MFDTSM method

to improve stock correlation analysis at both theoretical and practical levels. The empirical results demonstrate that MFDTSM performs effectively in industry correlation, linear correlation analysis, and stock correlation pricing, which validates its ability to handle complex datasets and proves its value for financial analysis and decision support.

## Related work

Early studies on stock correlation primarily employed classical statistical methods such as Manhattan Distance (MD) [20], Euclidean Distance (ED) [10], and Cosine Similarity (CS) [21]. These methods determine the similarity between stocks by measuring the geometric or spatial properties of time series [22]. For example, Zhu et al. [23] used an approach based on Manhattan Distance to estimate the similarity of state vectors at different time points in financial time series data, with the aim of estimating the embedding dimension of the financial time series. Esmalifalak [24] introduced an ED similarity measure that combines risk and return to provide a more comprehensive explanation of asset dependencies in financial markets. Additionally, Dong [25] developed a QUALIFLEX (Qualitative Flexible Multiple Criteria Method) based on CS for evaluating financial performance indicators. Although these methods provide a quantitative perspective for understanding the correlation of financial data, they focus primarily on a single-factor analysis and are unable to adequately capture the multidimensional complexity of market data.

Consequently, researchers have turned to more advanced time series analysis methods, such as Autoregressive Model (AR) [26], Autoregressive Integrated Moving Average (ARIMA) [27], and Generalized AutoRegressive Conditional Heteroskedasticity (GARCH) [28]. For example, Alam et al. [29] applied ARIMA models to study stock correlation and its application in portfolio optimization in the Dhaka Stock Exchange. These models utilize historical data on stock prices or returns to reveal dynamic interactions among stocks [30] [31]. To accurately handle time domain phase differences in time series data, methods such as the Longest Common Subsequence (LCSS) [14] and Dynamic Time Warping (DTW) [15] have been introduced. Soleimani et al. [32] proposed a method that combines LCSS with dual similarity thresholds, which significantly improves the accuracy and flexibility of stock correlation analysis. Li [33] developed a method combining DTW and time domain influence to optimize the similarity measure performance for financial time series data mining. Bai et al. [34] used DTW to analyze the financial network of the New York Stock Exchange to gain insight into the structural evolution of financial time series. Zhao et al. [35] proposed Dynamic Multi-Perspective Personalized Similarity Measurement (DMPSM), an improved framework based on the DTW algorithm, to enhance the interpretability of correlation among stock time series. Although these methods partially address the complex relationships and phase differences of time series, the effects of phase difference change rates or the multidimensionality of stock data on correlation analysis have not been fully explored.

In recent years, the research community in this field has begun to emphasize the importance of multidimensionality in stock data for understanding market dynamics [36]. Consequently, researchers have adopted advanced machine learning methods, including Principal Component Analysis (PCA) [37], Multivariate Empirical Mode Decomposition (MEMD) [38], and XGBoost [39] to analyze these complex datasets. Lettau et al. [40] evaluated potential asset price factors with the extended traditional PCA method, offering new perspectives for stock analysis multidimensionality. Huang and colleagues [41] applied the MEMD method to analyze a variety of factors influencing stock price volatility, comprehensively capturing the multidimensional characteristics of the stock market. Meanwhile, Vuong et al. [42] utilized XGBoost to select relevant stock factors, effectively integrating multidimensional data into stock market prediction models. Although these methods substantially improved the accuracy and efficiency of market analysis by comprehensively considering the multidimensional characteristics of stock data, they show limitations in capturing unique patterns of stock factors and the distinctive mechanisms behind each stock [43].

To address these limitations, our MFDTSM approach focuses on personalized analysis of the factors influencing patterns behind each stock and optimizes the results by quantifying the rate of time domain difference changes in time

series. This method successfully resolves the shortcomings of traditional analysis in capturing the multidimensionality and dynamics of stock data, thus improving the accuracy and efficiency of stock similarity measures.

## Method

The MFDTSM consists of three core modules. First, the factor influence quantification module employs an enhanced XGBoost model based on SHAP values to quantitatively analyze the influence of stock factors on market dynamics. Next, the cumulative distance matrix construction module integrates the factor weight vectors derived from the previous module to generate an optimal path cumulative distance matrix for similarity measurements. Finally, the TLE dynamics quantification module utilizes the cumulative distance matrix to quantify the rate of phase difference changes in TLE, dynamically adjusting the similarity measurement results to enhance the accuracy and robustness of MFDTSM.

## Quantification of factor influence

In this paper, we quantitatively analyze the multidimensionality of the stock data and propose an enhanced XGBoost model based on SHAP values. The model reveals how various stock factors reflect market dynamics, uncovering differences between factors and intrinsic connections of stock trends. To ensure consistency in the analysis, the selected stock datasets are consistent in terms of time span and factor set. Through preprocessing, each stock is transformed into multidimensional time series data, which is denoted by bold italicized capital letters $X$ in this article, i.e., $X = (X_1, X_2, \ldots, X_n)^\top = (X^1, X^2, \ldots, X^v) \in \mathbb{R}^{n \times v}$, where $n$ denotes the number of time points, and $v$ denotes the number of factors. In addition, $X$ corresponds to the label $y = (y_1, y_2, \ldots, y_n) \in \mathbb{R}^n$, which shows the closing price trend of the stock.

Based on this formulation, XGBoost is utilized to learn the multidimensional time series **X** that accurately predicts trends in stock prices **y**. The optimization process involves minimizing the objective function $L(\theta)$ to find a balance between model prediction accuracy and complexity:

$$L(\theta) = \sum_{i=1}^{n} l(y_i, \hat{y}_i(\mathbf{X}_i)) + \sum_{k=1}^{K} \Omega(f_k) \tag{1}$$

where $\hat{y}_i(\mathbf{X}_i)$ denotes the predicted value of sample $\mathbf{X}_i \in \mathbb{R}^v$:

$$\hat{y}_i(\mathbf{X}_i) = \sum_{k=1}^{K} f_k(\mathbf{X}_i) \tag{2}$$

where, $f_k(\mathbf{X}_i)$ is the $k$-th gradient boosting tree for the $i$-th predicted value of the sample. In addition, $l$ is defined as the Mean Square Error (MSE), which measures the prediction error:

$$l(y_i, \hat{y}_i(\mathbf{X}_i)) = (y_i - \hat{y}_i(\mathbf{X}_i))^2 \tag{3}$$

while the regularization term $\Omega(f_k)$ is defined as:

$$\Omega(f_k) = \gamma T_k + \frac{1}{2}\lambda \|w_k\|^2 \tag{4}$$

where $T_k$ and $w_k$ are the number of leaves and leaf weights of the $k$-th tree, respectively. The hyperparameters $\gamma$ and $\lambda$ control the complexity of the number of leaves and the weights of L2 regularization.

After model training, SHAP is used to quantitatively analyze and explain each factor's influence on stock price trends. SHAP values, as an advanced explanatory model, not only accurately quantify the influence of each factor in stock market dynamics but also reveal the interactions between different factors. For sample $i$ and factor $j$, the SHAP value $\phi_{ij}$ is calculated as follows:

$$\phi_{ij} = \sum_{S \subseteq F \setminus \{j\}} \frac{|S|!(|F| - |S| - 1)!}{|F|!} \left[ \hat{y}_i(\mathbf{X}_i) - \hat{y}_i(\mathbf{X}_i^S) \right] \tag{5}$$

where $F$ denotes the full set of factors, and $S$ is the subset of factors excluding $j$. The symbols $|S|$ and $|F|$ represent the number of elements in sets $S$ and $F$, respectively, while $\hat{y}_i(\mathbf{X}_i^S)$ refers to the predicted value of sample $i$ under the influence of subset $S$. Further, by aggregating and normalizing the SHAP values of the factors, we obtain the vector of SHAP values $\phi \in \mathbb{R}^v$, where $\phi_j$ denotes the value of the $j$-th element in the vector:

$$\phi_j = \frac{|\sum_{i=1}^{n} \phi_{ij}|}{\sum_{k=1}^{v} |\sum_{i=1}^{n} \phi_{ik}|} \tag{6}$$

where the vector of SHAP values $\phi = (\phi_1, \phi_2, \ldots, \phi_v)$ reflects the relative influence of each factor on the stock price dynamics.

Following this, the K-means clustering algorithm [43] is employed to group all stocks, aiming to identify stocks with similar factor influence patterns and classifying them into the same group. During the clustering process, the centroids are initialized based on the Euclidean distances of SHAP value vectors among the stocks. The positions of the centroids are then optimized through an iterative process to minimize the sum of the distances of the stocks within each cluster to their centroids until a predefined number of clusters $c$ is reached.

To further analyze the commonalities within each cluster, the average SHAP value vectors of all stocks in the cluster are calculated, defining a common set of factor weights for each cluster $\mathbf{W}^i \in \mathbb{R}^v$:

$$\mathbf{W}^i = \frac{1}{m_i} \sum_{j=1}^{m_i} \phi^{(i,j)} \tag{7}$$

where $m_i$ represents the number of stocks in cluster $i$, and $\phi^{(i,j)} \in \mathbb{R}^v$ represents the SHAP value vector of the $j$-th stock in the cluster. The above calculation yields the factor weight matrix $\mathbf{W} = (\mathbf{W}^1, \mathbf{W}^2, \ldots, \mathbf{W}^c)^\top \in \mathbb{R}^{c \times v}$, where $\mathbf{W}^k$, $k \in [1, c]$, represents the weights of the stock factors in cluster $k$, which ensures a consistent evaluation of factor influence within clusters, facilitating the understanding of shared characteristics and market behavior among stocks within the cluster.

## Cumulative distance matrix construction

To accurately measure the similarity between stock time series while fully incorporating their multidimensional characteristics, our MFDTSM method introduces the DTW technique by constructing a cumulative distance matrix [18]. In this section, two multidimensional time series of stocks $\mathbf{X} \in \mathbb{R}^{n \times v}$ and $\mathbf{Y} \in \mathbb{R}^{n \times v}$, belonging to the same cluster, are used as an example to explain the process of constructing the cumulative distance matrix, where $\mathbf{W}^k \in \mathbb{R}^v$ is the factor weight vector of the stocks in the cluster.

Before constructing the cumulative distance matrix, a weighted Euclidean distance $d^w$ is defined to measure the distance between two multidimensional data points. In $d^w$, the weight vectors of the stock factors $\mathbf{W}^k$ are considered, which enables the distance measure to reflect the extent to which different factors contribute to stock similarity. The definition of

the distance $d^w$ is as follows:

$$d^w_{ij} = |\mathbf{W}^k \cdot \mathbf{X}_i - \mathbf{W}^k \cdot \mathbf{Y}_j| \tag{8}$$

In this formula, $d^w_{ij}$ is defined as the weighted difference between $v$ dimensional data points $\mathbf{X}_i$ and $\mathbf{Y}_j$ at respective time points $i$ and $j$, with $\mathbf{W}^k$ as the factor weight vector.

Subsequently, based on $d^w$, the cumulative distance matrix $\mathbf{M} \in \mathbb{R}^{n \times n}$ is constructed, which records the cumulative distances across all possible alignments between two time series. The cumulative distance matrix $\mathbf{M}$ not only enhances the flexibility of the similarity measure but also improves the accuracy. The elements $M_{ij}$ in $\mathbf{M}$ represent the minimum cumulative distance from the start of the sequence to the $(i,j)$ position, which is calculated as follows:

$$M_{i,j} = \begin{cases} d^w_{1,1} & i = 1 \text{ and } j = 1, \\ M_{i-1,j} + d^w_{i,j} & i > 1 \text{ and } j = 1, \\ M_{i,j-1} + d^w_{i,j} & i = 1 \text{ and } j > 1, \\ \min\left(M_{i,j-1}, M_{i-1,j}, M_{i-1,j-1}\right) + d^w_{i,j} & \text{otherwise.} \end{cases} \tag{9}$$

this computation ensures that the similarity measure $\mathbf{M}$ is based not only on the static characteristics of the time series but also considers dynamic changes, providing a comprehensive basis for similarity assessment. The final cumulative distance matrix $\mathbf{M}$ is constructed as follows:

$$\mathbf{M} = \begin{bmatrix} M_{1,1} & \cdots & M_{1,n} \\ \vdots & \ddots & \vdots \\ M_{n,1} & \cdots & M_{n,n} \end{bmatrix} \tag{10}$$

where $M_{n,n}$ as the final element in the matrix, represents the cumulative distance of the globally optimal path between two time series, becoming the key indicator for evaluating the similarity between $\mathbf{X}$ and $\mathbf{Y}$.

## Quantification of TLE dynamics

This section details the process of quantifying the dynamics of phase differences in the stock market's TLE. The MFDTSM method analyzes the cumulative distance matrix $\mathbf{M}$ and quantifies the dynamic phase differences between stocks, adjusting the similarity measure result $M_{n,n}$ accordingly to improve accuracy and robustness.

First, the global optimal path is extracted from the distance matrix $\mathbf{M}$. The procedure is as follows: starting from the lower right corner of matrix $\mathbf{M}$, i.e., the $(n,n)$ position, and gradually tracing back to the upper left corner $(1,1)$. In this process, each step of the path $P_k$ is defined by the following equation:

$$P_k = (i_k, j_k) = \text{argmin}(f_k(i_{k-1}, j_{k-1})) \tag{11}$$

where function argmin determines the coordinate point $(i_k, j_k)$ that minimizes $f_k(i_{k-1}, j_{k-1})$. The function $f_k(i,j)$ then selects the minimum cumulative distance among the three neighboring points of $(i,j)$:

$$f_k(i,j) = \min\left(M_{i-1,j}, M_{i,j-1}, M_{i-1,j-1}\right) \tag{12}$$

and through this iterative process, the vector representing the globally optimal path is formed as $\mathbf{P} = (P_1, P_2, \ldots, P_L)$, where $L$ is the total length of the path.

Furthermore, we analyze the path **P**, which reveals key characteristics of phase difference dynamics. The directional changes between adjacent coordinate points reflect the rate of phase difference changes between stocks, geometrically manifested as turning points in the path **P**. Specifically, a point $P_i$ is defined as a turning point when its directional difference with the subsequent point $P_{i+1} - P_i$ differs from that with its preceding point $P_i - P_{i-1}$. Based on this, we propose a method to accurately calculate $L'$, the total number of turning points in path **P**, calculated as follows:

$$L' = \sum_{i=2}^{L-1} \mathbb{I}(P_{i+1} - P_i \neq P_i - P_{i-1}) \tag{13}$$

where $\mathbb{I}$ is an indicator function that determines whether each point meets the conditions of the turning point. The function value is 1 when the condition is established; otherwise, it is 0. With $L'$ as the total number of turning points, the phase change rate is defined as:

$$r = \frac{L'}{L - 2} \tag{14}$$

where rate of change $r$ varies between values [0,1], which directly reflects the degree of dynamic change in phase relationships between time series.

Finally, combining the rate $r$ and the original cumulative distance $M_{n,n}$, the similarity measurement result is optimized to improve its accuracy and adaptability:

$$\text{MFDTSM}(\mathbf{X}, \mathbf{Y}) = M_{n,n} \cdot (1 + \mu r) \tag{15}$$

where $\mu$ is the adjustment coefficient used to balance the influence of the phase change rate $r$ on the result, ensuring the comprehensiveness and stability of the measurement.

## Experiments and analysis

This chapter aims to provide an experimental assessment of the MFDTSM method. We constructed a comprehensive evaluation framework, focusing on verifying the effectiveness and robustness of the MFDTSM method in the field of multidimensional time series similarity measurement. This section systematically introduces the dataset, experimental setup, and an in-depth analysis of the experimental results.

### Experimental dataset

In this study, the utilized data is from the JoinQuant Quantitative Research Platform, covering 102 stocks from the telecommunications and financial industries during the period from January 1, 2021, to December 31, 2022. The analysis involves 12 critical stock factors, including price, turnover rate, and moving averages. All selected stocks are categorized according to the Shenwan Industry Classification Standard. The dataset in the study is divided into a training set and a test set in a 7:3 ratio. To ensure the consistency and reliability of the data, thorough data cleaning and normalization were performed to eliminate potential biases and anomalies. The normalization formula is as follows:

$$X'_i = \frac{X_i - X_{i-1}}{X_{i-1}} \tag{16}$$

where $X_i$ represents the original stock data for the $i$-th day, and $X'_i$ represents the processed data.

## Experimental setup

The experimental parameters of the MFDTSM method were carefully configured to ensure model performance and accuracy. The XGBoost model performs a short-term price prediction for the closing price on the next day of each stock, where the number of boosting rounds $K$ is set to 100, the minimum loss reduction of leaf nodes $\gamma$ is 0, and the L2 regularization term of weights $\lambda$ is set to 1 to prevent overfitting and maintain generalization capability. The learning rate and the maximum depth of the trees are set to 0.3 and 6, respectively, to optimize the learning speed and complexity. The selection of hyperparameters, including the number of clusters $c = 5$ and the phase change rate coefficient $\mu = 1$, was determined through a grid search and cross-validation process to ensure robustness. We evaluated clustering performance using the Silhouette Score and Elbow Method, which indicated that $c = 5$ optimally balances intra-cluster cohesion and inter-cluster separation. Similarly, $\mu$ was tuned in the range [0.5,2.0] to minimize prediction error in a validation set. Although the model performance is relatively stable across slight variations in these parameters, future work could incorporate Bayesian optimization for more efficient hyperparameter tuning.

The experimental evaluation methodology covers the following aspects: firstly, the correlation between industry relationships and the similarity between stocks was analyzed by calculating the similarity between stocks in different industries;secondly, PCC were used to validate the stability of the proposed methodology, which assesses the validity of the similarity metric by analyzing the linear relationships between pairs of stocks; finally, the methodology's interpretability and practical value were assessed by applying it to stock pricing based on their similarity to other stocks.

## Methods of comparison

As shown in Table 1, the proposed MFDTSM method was compared with various similarity measurement methods to comprehensively evaluate its performance. These comparison methods included traditional benchmark similarity measures, such as ED and DTW, as well as other advanced stock data analysis methods like LCSS and TDTW. Additionally, two ablation study methods were considered: MFDTSM-M, which removes the factor influence quantification module from MFDTSM, and MFDTSM-T, which removes the TLE dynamic quantification module. This comparative analysis helps reveal the importance of each component in MFDTSM and its impact on overall performance.

## Results analysis and summary

This section provides a comprehensive experimental evaluation of the effectiveness of the MFDTSM method, focusing on its application in multidimensional time series data similarity measurement. Particularly in stock data analysis, MFDTSM demonstrates significant superiority and stability.

**Industry relationship analysis.** This experiment explores into the industry relevance by different methods in identifying stock similarities and evaluates the performance of each method in calculating stock similarities by comparing the

**Table 1**. Experimental comparison methods.

| Similarity Metric | Description |
|---|---|
| ED | Euclidean Distance |
| DTW | Dynamic Time Warping |
| LCSS | Longest Common Subsequence |
| TDTW | DTW extension method based on time domain weights [33] |
| DMPSM | A Dynamic Multi-Perspective Personalized Similarity Measurement Approach [35] |
| MFDTSM | Multi-dimensional dynamic time-domain similarity metrics |
| MFDTSM-M | MFDTSM without the factor influence quantification module |
| MFDTSM-T | MFDTSM without the temporal TLE dynamics quantization module |

probability of whether a stock pair belongs to the same primary or secondary industries. As shown in Fig 1, the experimental results demonstrate the variation in probabilities for the top 20 to 200 most similar stock pairs identified by the different methods, in terms of primary and secondary industries. The experimental results show that, in the primary industry relationship analysis, MFDTSM gradually outperforms all other methods as the number of pairs of stocks increases. Notably, when the number of pairs of stocks reaches 200, MFDTSM identifies stocks from the same primary industry with an accuracy of 88.5%, approximately 6% higher than ED, while MFDTSM-T achieves an accuracy of 84.5%. This significant performance improvement is attributed to the factor influence quantization module in MFDTSM, which dynamically explains the impact of different factors on stock trends and improves the accuracy of the similarity measure.

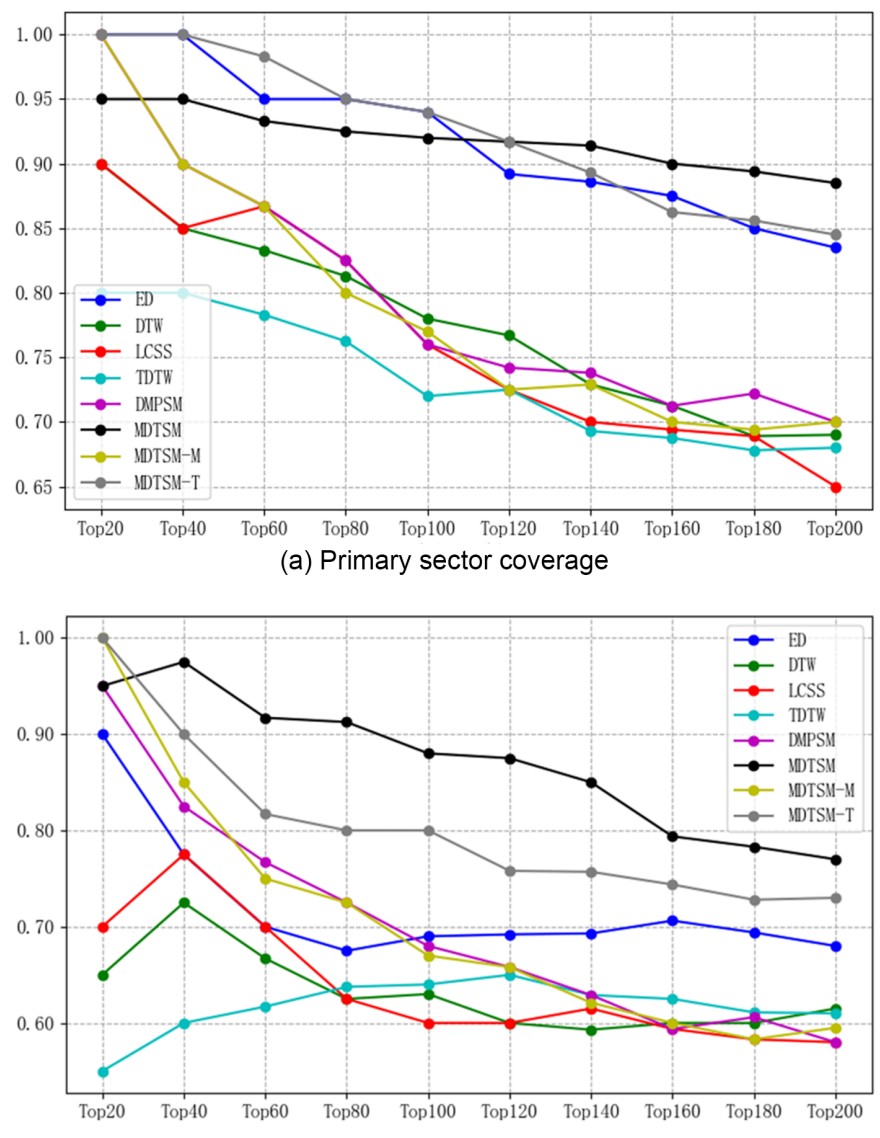

(a) Primary sector coverage

(b) Secondary sector coverage

**Fig 1**. **Comparison of the performance of different methods for recognizing stock industry relationships.**

Furthermore, the TLE dynamic quantification module improves the method's stability, ensuring that MFDTSM continues to lead in performance.

In the secondary industry analysis, all methods show increased volatility due to the more detailed classification. However, MFDTSM maintains a lead of at least 12% compared to other benchmarking methods. It is worth noting that the ablation variants of MFDTSM (MFDTSM-M and MFDTSM-T) are ahead of those of MFDTSM at the beginning of the experiments, but their performance decreases significantly as the experiments progress, especially for MFDTSM-M, highlighting the central role and importance of multi-factor fusion methods in the MFDTSM framework. The comprehensive analysis shows that MFDTSM outperforms other methods in industry relationship analysis by more than 10% on average, demonstrating its advantage in the field of multidimensional time series similarity measurement.

**Linear correlation analysis.** This study aims to evaluate the effectiveness of different similarity measures in financial time series analysis, specifically focusing on their capability to capture linear correlations between stocks. The experiment measures the linear relationship between pairs of stocks by employing PCC as a standard statistical tool to reflect the strength of linear correlation between stock price movements in the test set. The results shown in Table 2 indicate that the 200 most similar stock pairs identified by different methods are different in means, and the larger values indicate closer linear correlation, which in turn reflects the accuracy and usefulness of the similarity measures.

The results highlight the superior performance of MFDTSM and its variant MFDTSM-T among all the compared methods, attributed to the integrated consideration of multiple stock factors in the MFDTSM approach, especially the key role played by the multidimensionality analysis of the factor impact quantification module in improving the computational accuracy. As the number of stock pairs increases, the performance difference between MFDTSM and MFDTSM-T becomes more pronounced, emphasizing the contribution of the TLE dynamic quantification module in analyzing phase difference change rates for algorithm stability. Compared to all other methods, ED performs second best, with a PCC of 0.679 for the top 200 stock pairs. However, MFDTSM outperforms ED by approximately 16%, demonstrating its capability to uncover linear relationships in stock data and its efficiency and stability in dealing with similarity measures of multidimensional time series data.

## Equity correlation pricing

To evaluate MFDTSM's performance in stock correlation pricing and to compare the performance of different methods in this area, this experiment uses comparative analysis. First, stocks are ranked according to their most similar stocks and ranked according to the average similarity of their most similar stocks, and the top-ranked stocks are selected as the study object. Multiple machine learning algorithms, including Linear Regression (LR), Random Forest (RF), and Support Vector Machine (SVM), are employed in this study to predict stock prices using the closing price of the target stock and its top $s$ similar stocks. In this experiment, $s$ and $n$ are set to 10, respectively, to evaluate the model by the metric: the difference between the actual closing price $Y$ and the predicted closing price $\hat{Y}$, where $\hat{Y}$ is derived from the analysis of $s$ similar stocks.

As Fig 2 shows, MFDTSM's average $R^2$ values under LR, RF, and SVM models outperformed other similarity measurement methods, particularly excelling in the RF model. This result indicates MFDTSM's significant advantage in handling

**Table 2**. Comparison of mean values of PCC for different methods.

| Method | ED | DTW | LCSS | TDTW | DMPSM | MFDTSM | MFDTSM-M | MFDTSM-T |
|---|---|---|---|---|---|---|---|---|
| Top 50 | 0.854 | 0.766 | 0.761 | 0.696 | 0.776 | 0.864 | 0.765 | 0.877 |
| Top 100 | 0.770 | 0.633 | 0.683 | 0.607 | 0.660 | 0.839 | 0.686 | 0.838 |
| Top 150 | 0.722 | 0.613 | 0.630 | 0.589 | 0.633 | 0.814 | 0.639 | 0.795 |
| Top 200 | 0.679 | 0.592 | 0.587 | 0.591 | 0.619 | 0.791 | 0.632 | 0.758 |

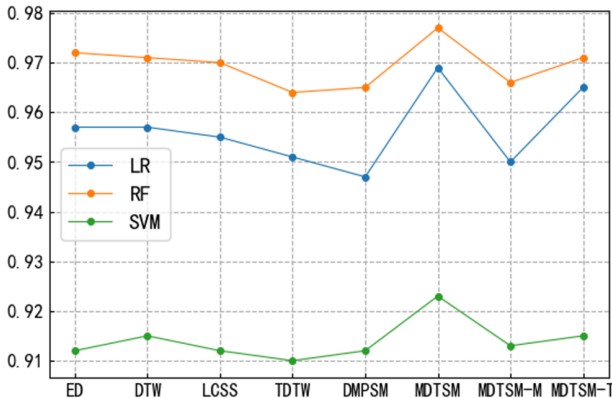

**Fig 2**. $R_2$ scores of different methods in machine learning models.

**Table 3**. Performance of different methods under machine learning models.

| | ED | DTW | LCSS | TDTW | DMPSM | MFDTSM | MFDTSM-M | MFDTSM-T |
|---|---|---|---|---|---|---|---|---|
| **MAE** | | | | | | | | |
| LR | 0.0274 | 0.0280 | 0.0308 | 0.0314 | 0.0306 | 0.0245 | 0.0305 | 0.0262 |
| RF | 0.0201 | 0.0206 | 0.0211 | 0.0234 | 0.0219 | 0.0189 | 0.0220 | 0.0201 |
| SVM | 0.0457 | 0.0435 | 0.0452 | 0.0465 | 0.0445 | 0.0412 | 0.0445 | 0.0428 |
| **MAPE** | | | | | | | | |
| LR | 0.0839 | 0.0863 | 0.0903 | 0.1088 | 0.0767 | 0.0717 | 0.0768 | 0.0710 |
| RF | 0.0514 | 0.0546 | 0.0548 | 0.0674 | 0.0525 | 0.0481 | 0.0523 | 0.0522 |
| SVM | 0.1543 | 0.1398 | 0.1531 | 0.1729 | 0.1195 | 0.1191 | 0.1205 | 0.1326 |
| **MEAN VALUE** | 0.0638 | 0.0621 | 0.0659 | 0.0751 | 0.0576 | 0.0539 | 0.0578 | 0.0575 |

complex features and correlations. The excellent performance of MFDTSM in each model further validates its efficiency in multidimensional time series similarity analysis.

Table 3 shows the mean values of MAE and MAPE for different methods under LR, RF, and SVM models. Fig 3, on the other hand, shows the distribution of MAE and MAPE of these methods under LR, RF, and SVM models in the form of box plots. In these box plots, the boxes show the interquartile range (IQR), spanning from the 25th to the 75th percentiles of the prediction errors. The horizontal lines inside the boxes denote the medians, while the whiskers extend to the maximum and minimum values within 1.5 times the IQR. Any data points beyond the whiskers are plotted as individual outliers. This visualization facilitates an intuitive comparison of both the central tendency, as reflected by the medians, and the dispersion of prediction errors across the different methods. The data indicate that MFDTSM outperforms the other techniques in terms of prediction accuracy and error rate for all test models. Notably, MFDTSM demonstrates the smallest median and narrower distribution range in MAE, indicating higher consistency and reliability. Although it didn't surpass DMPSM in MAPE performance under the SVM model, MFDTSM still performed best in LR and RF models. Additionally, MFDTSM-T's performance followed closely, highlighting the importance of the factor influence quantification module. Overall, MFDTSM showed an average lead of about 5% in time series similarity measurement, demonstrating its comprehensive superiority.

## Conclusions

This paper develops an innovative stock similarity measurement method, MFDTSM, to comprehensively and accurately evaluate relationships between stocks and provide reliable decision-making support for investors. Experimental analyses demonstrate the superior performance of MFDTSM across multiple dimensions. In industry relationship analysis,

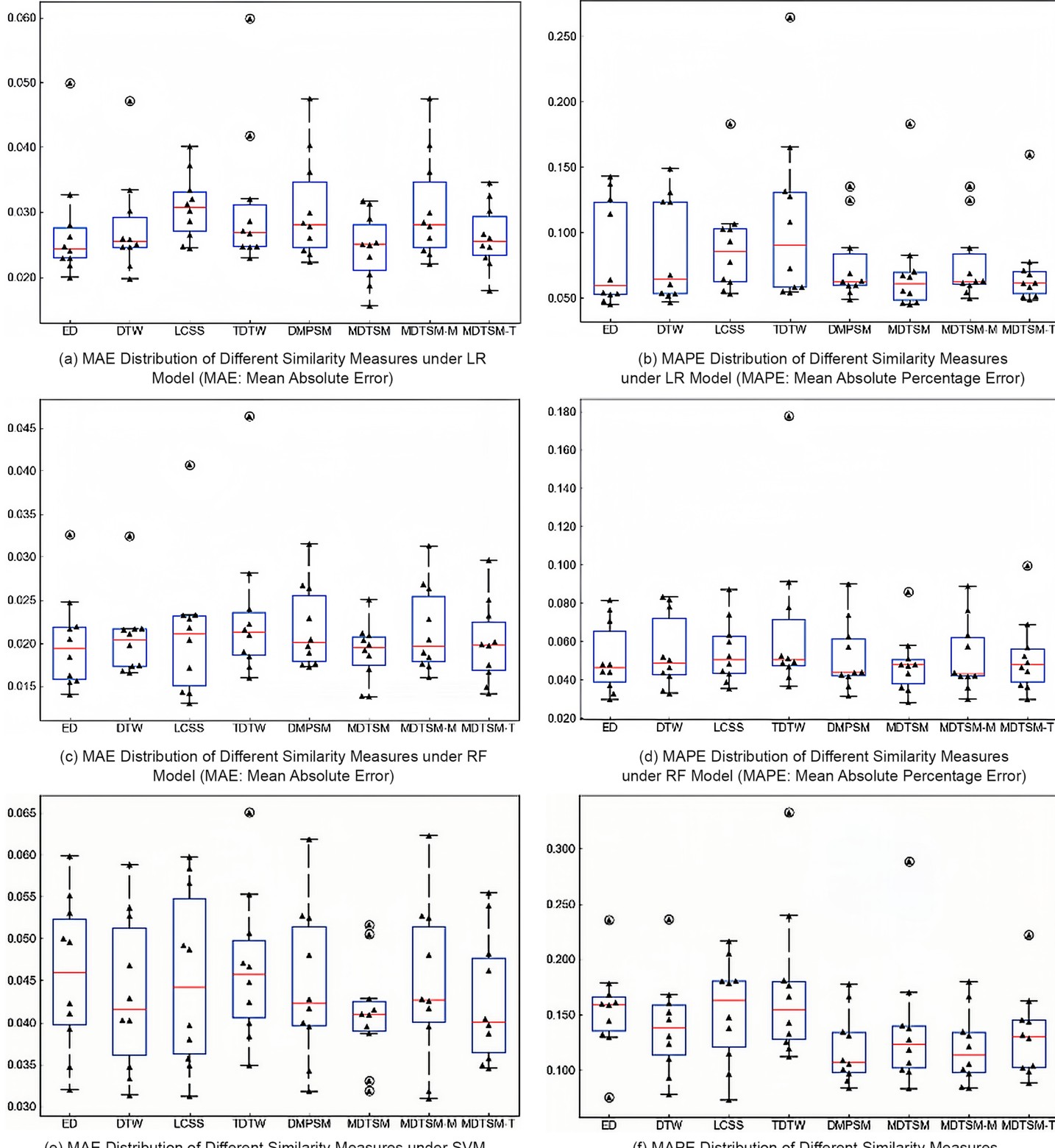

(a) MAE Distribution of Different Similarity Measures under LR Model (MAE: Mean Absolute Error)

(b) MAPE Distribution of Different Similarity Measures under LR Model (MAPE: Mean Absolute Percentage Error)

(c) MAE Distribution of Different Similarity Measures under RF Model (MAE: Mean Absolute Error)

(d) MAPE Distribution of Different Similarity Measures under RF Model (MAPE: Mean Absolute Percentage Error)

(e) MAE Distribution of Different Similarity Measures under SVM Model (MAE: Mean Absolute Error)

(f) MAPE Distribution of Different Similarity Measures under SVM Model (MAPE: Mean Absolute Percentage Error)

**Fig 3**. **Box Plot of Evaluation Parameters (MAE, MAPE) for Different Similarity Measures in Machine Learning Models (LR, RF, SVM).**

MFDTSM identifies stocks in the same industry with exceptional accuracy. In linear correlation analysis, MFDTSM exhibits superior capability in detecting linear relationships in the test set, highlighting its accuracy and stability. In stock correlation pricing analysis, MFDTSM surpasses comparative methods across key metrics including $R^2$, MAE, and MAPE, showing its superiority in stock price analyzing ability. Moreover, MFDTSM demonstrates superior overall performance compared to its ablation method variants, validating the importance of the factor impact quantification and TLE dynamics quantification modules. These results confirm that incorporating the multidimensionality of stocks and the dynamics of TLE can improve the accuracy and stability of similarity computation. Furthermore, the selection of the 12 stock factors (e.g., price, turnover rate, moving averages) was based on both the established financial research consensus and preliminary feature importance analysis using tree-based models. These factors are widely recognized as key indicators of stock performance and market behavior. To assess the stability of MFDTSM with respect to factor selection, we conducted sensitivity analyses by alternately excluding one factor at a time. The results showed that the model maintains robust performance with minimal fluctuation in similarity accuracy (±2.3% in Mean Absolute Percentage Error, MAPE), indicating that MFDTSM is not overly reliant on any single factor and is generally stable under factor variation.

Despite its impressive performance, MFDTSM presents opportunities for further enhancement. Its hyperparameter selection has not been extensively experimentally tested, and future research needs to explore it in depth to determine the optimal performance parameters. Furthermore, although MFDTSM reduces the number of similarity calculations by clustering stocks, its time complexity is still high. Future research should focus on optimizing the algorithm's operational efficiency and reducing the consumption of computational resources while improving the accuracy.

## Supporting information

**S1 Table. Complete list of 102 research stocks.** This Excel spreadsheet contains the complete list of 102 research stocks analyzed in this study. The data include:

1) Stock Code;
2) Stock Name;
3) Primary Industry (L1) based on Shenwan Industry Classification;
4) Secondary Industry (L2) based on Shenwan Industry Classification.

The full dataset is available as S1 Table.

(XLSX)

## Acknowledgments

Project Supported by the Science Research Foundation for Middle-Aged and Young Scholars of Qinghai Normal University (Grant No. 2023QZR001).

## Author contributions

**Conceptualization:** Jinyu Fan, Guanyu Lu, Jun Ma.

**Data curation:** Jinyu Fan, Guanyu Lu, Jun Ma.

**Formal analysis:** Jinyu Fan, Guanyu Lu, Jun Ma.

**Investigation:** Jinyu Fan, Guanyu Lu, Jun Ma.

**Methodology:** Jinyu Fan, Guanyu Lu, Jun Ma.

**Project administration:** Jinyu Fan, Jun Ma.

**Resources:** Jinyu Fan, Jun Ma.

**Supervision:** Jinyu Fan, Guanyu Lu.

**Validation:** Jun Ma.

**Visualization:** Guanyu Lu.

**Writing – original draft:** Jinyu Fan, Jun Ma.

**Writing – review & editing:** Guanyu Lu, Jun Ma.

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
