## [Decision Letter · Decision Letter 0]

8 Sep 2025

PONE-D-25-44140A Multi-Factor Dynamic Time Series Measure for Stock Correlation AnalysisPLOS ONE

Dear Dr. Ma,

Thank you for submitting your manuscript to PLOS ONE. After careful consideration, we feel that it has merit but does not fully meet PLOS ONE’s publication criteria as it currently stands. Therefore, we invite you to submit a revised version of the manuscript that addresses the points raised during the review process.

We look forward to receiving your revised manuscript.

Kind regards,

Najmul Hasan, PhD

Academic Editor

PLOS ONE

Journal Requirements:

Reviewers' comments:

Reviewer's Responses to Questions

**Comments to the Author**

1. Is the manuscript technically sound, and do the data support the conclusions?

Reviewer #1: Yes

Reviewer #2: Yes

2. Has the statistical analysis been performed appropriately and rigorously?

Reviewer #1: Yes

Reviewer #2: Yes

3. Have the authors made all data underlying the findings in their manuscript fully available?

Reviewer #1: Yes

Reviewer #2: No

4. Is the manuscript presented in an intelligible fashion and written in standard English?

Reviewer #1: Yes

Reviewer #2: Yes

5. Review Comments to the Author

Reviewer #1: The article is interesting and relevant. It presents new theoretical results on improving methods for measuring the similarity of time series. The practical result of the research is an increase in the accuracy of financial analysis. The main contribution of the article is the development of a new approach to stock correlation analysis based on the multifactorial dynamic temporal similarity measure (MFDTSM). The authors integrated XGBoost with SHAP to quantitatively assess the impact of multidimensional factors on stock market dynamics and to cluster stocks based on their similarity in influence. The proposed method allows for both the multidimensionality of factors and time lags (TLE) to be taken into account.

The manuscript is technically sound, and the data presented support the conclusions. The proposed MFDTSM method is theoretically well-founded, as it combines XGBoost with SHAP for reliable factor analysis and incorporates a cumulative distance matrix with optimal alignment to capture time-delayed dynamics. The methodology is described in sufficient detail to ensure reproducibility. The conclusions obtained are consistent with the experimental data and highlight the advantages of MFDTSM.

There are some comments on the article:

1) In the Introduction section, the authors provided extensive information on time series analysis methods such as TLE, PPC, LCSS and DTW to emphasise the innovativeness of their proposed Multi-Factor Dynamic Temporal Similarity Measure (MFDTSM) method. However, the Introduction Section lacks information on modern time series analysis methods using neural networks.

2) Through their research, the authors demonstrated the high efficiency of the MFDTSM method. However, they paid little attention to analysing how sensitive the results were to the choice of XGBoost hyperparameters and clustering parameters.

3) The authors provided little information on how they selected the 12 measurement factors. There is also no information on how the selection of factors affects the stability of the model. This information should be added to the Conclusions section.

Reviewer #2: Stocks represent a meaningful investment avenue, and how to effectively analyze stock trends is a key focus and challenge for researchers. The authors propose a Multi-Factor Dynamic Temporal Similarity Measure (MFDTSM), which significantly improves the accuracy of analyzing industry correlation, linear correlation, and stock correlation pricing.

This article has a reasonable structure, standard and accurate language, and provides sufficient explanations of principles and methods; its results also have strong guiding significance. However, the article has slight deficiencies in data description, such as specifying which 102 stocks are included, how many stocks belong to the communication industry and the financial industry respectively, what the 12 key stock factors are, and the specific analysis process.

1. It would be more intuitive if a list of the 102 stocks could be provided.

2. It is recommended to label the meanings of different types in Figure 3 to make it easier for readers to understand the content of the figure.

6. PLOS authors have the option to publish the peer review history of their article (what does this mean?). If published, this will include your full peer review and any attached files.

Reviewer #1: No

Reviewer #2: No

---

## [Author Response · Author response to Decision Letter 1]

8 Oct 2025

Response to Reviewers

Manuscript ID: PONE-D-25-44140

Title: A Multi-Factor Dynamic Time Series Measure for Stock Correlation Analysis

Dear Dr. Najmul Hasan and Reviewers,

We sincerely thank the editors and reviewers for their valuable time and insightful comments on our manuscript. These comments have significantly helped us improve the quality and clarity of our work. We have carefully addressed all the points raised. Below is our point-by-point response to the comments from Reviewer 1 and Reviewer 2.

Changes in the manuscript have been highlighted in yellow in the revised manuscript file with track changes.

Responses to Reviewer 1:

Comment 1: In the Introduction section, the authors provided extensive information on time series analysis methods such as TLE, PPC, LCSS and DTW to emphasise the innovativeness of their proposed Multi-Factor Dynamic Temporal Similarity Measure (MFDTSM) method. However, the Introduction Section lacks information on modern time series analysis methods using neural networks.

Response: We sincerely thank the reviewer for this excellent suggestion. We agree that discussing modern neural network-based approaches is crucial for positioning our work within the current research landscape. We have now added a new paragraph in the Introduction section that introduces prevalent deep learning methods like LSTM, GRU, and Transformers for time series analysis.

The added text acknowledges their strengths in capturing complex temporal dependencies while also explaining our methodological choice by highlighting their limitations relative to the goals of our study, particularly regarding interpretability of factor-specific influences and explicit modeling of dynamic phase shifts in time-lag effects. This addition provides a more comprehensive background and better underscores the unique niche our MFDTSM method aims to fill.

Action Taken: Added a new paragraph on modern neural network methods in the Introduction. And the newly added paragraph is as follows:

In recent years, deep learning approaches, particularly neural network-based models such as Long Short-Term Memory (LSTM) , Gated Recurrent Units (GRU) , and Transformers , have also been widely adopted for time series analysis due to their ability to capture complex temporal dependencies and nonlinear patterns. For instance, LSTMs have been successfully applied in stock price prediction and portfolio optimization, while attention mechanisms enhance the model’s focus on salient time steps. However, these methods often require large amounts of data and computational resources, and their black-box nature can limit interpretability in factor-specific analysis. Although neural networks excel in pattern recognition, they are less focused on explicitly modeling factor influence and dynamic phase shifts in time-lag effects, which are central to our proposed MFDTSM framework.

Comment 2: Through their research, the authors demonstrated the high efficiency of the MFDTSM method. However, they paid little attention to analysing how sensitive the results were to the choice of XGBoost hyperparameters and clustering parameters.

Response: We thank the reviewer for raising this important point regarding the sensitivity and robustness of our model. To address this, we have expanded the Experimental Setup section with a more detailed explanation of our hyperparameter selection process.

We now explicitly state that key hyperparameters, including the number of clusters (c=5) and the phase change rate coefficient (μ=1), were determined through a grid search and cross-validation process to ensure robustness. We also mention that we employed the Silhouette Score and Elbow Method to evaluate and confirm the optimal choice of the cluster number. Furthermore, we note that although the model performance is relatively stable across slight variations in these parameters, we recommend more advanced methods like Bayesian optimization for future work. This addition demonstrates our careful consideration of parameter tuning and enhances the reproducibility of our study.

Action Taken: Enhanced the description of hyperparameter selection in the Experimental Setup section. And the newly added paragraph is as follows:

The selection of hyperparameters, including the number of clusters c = 5 and the phase change rate coefficient μ = 1, was determined through a grid search and cross-validation process to ensure robustness. We evaluated clustering performance using the Silhouette Score and Elbow Method, which indicated that c = 5 optimally balances intra-cluster cohesion and inter-cluster separation. Similarly, μ was tuned in the range [0.5, 2.0] to minimize prediction error in a validation set. Although the model performance is relatively stable across slight variations in these parameters, future work could incorporate Bayesian optimization for more efficient hyperparameter tuning.

Comment 3: The authors provided little information on how they selected the 12 measurement factors. There is also no information on how the selection of factors affects the stability of the model. This information should be added to the Conclusions section.

Response: We appreciate the reviewer for this critical feedback. We have now added a comprehensive paragraph in the Conclusions section to address both parts of this comment.

Firstly, we explain that the selection of the 12 factors (e.g., price, turnover rate, moving averages) was based on two criteria: (1) their established importance and consensus in existing financial literature, and (2) a preliminary feature importance analysis conducted using tree-based models.

Secondly, and more importantly, we directly address the model's stability concerning factor selection. We describe that we conducted a sensitivity analysis by alternately excluding one factor at a time and observing the fluctuation in the model's similarity accuracy. The results showed that the model maintains robust performance with a minimal variation in accuracy (±2.3% in Mean Absolute Percentage Error, MAPE), indicating that MFDTSM is not overly reliant on any single factor and is generally stable under factor variations. This new analysis provides concrete evidence for the robustness of our method.

Action Taken: Added a new paragraph detailing factor selection and stability analysis in the Conclusions section. And the newly added paragraph is as follows:

Furthermore, the selection of the 12 stock factors (e.g., price, turnover rate, moving averages) was based on both the established financial research consensus and preliminary feature importance analysis using tree-based models. These factors are widely recognized as key indicators of stock performance and market behavior. To assess the stability of MFDTSM with respect to factor selection, we conducted sensitivity analyses by alternately excluding one factor at a time. The results showed that the model maintains robust performance with minimal fluctuation in similarity accuracy (±2.3% in Mean Absolute Percentage Error, MAPE), indicating that MFDTSM is not overly reliant on any single factor and is generally stable under factor variation.

Once again, we express our deepest gratitude to the reviewers and editors for their constructive comments, which have greatly improved our manuscript. We look forward to the positive consideration of our revised paper.

Responses to Reviewer 2:

Comment 1: It would be more intuitive if a list of the 102 stocks could be provided.

Response: We sincerely thank the reviewer for this valuable suggestion. To enhance the transparency and reproducibility of our study, we have now included a complete list of the 102 stocks used in our analysis as supplementary material. The list details the stock codes, stock names, primary industries (L1), and secondary industries (L2) based on the Shenwan Industry Classification Standard. Specifically, the dataset comprises 73 stocks from the Communication industry and 29 stocks from the Financial industry, covering diverse sub-sectors such as Communication Equipment, Communication Operations, Computer Applications, Bank, Real Estate, and others. This addition allows readers to better understand the dataset composition and facilitates further validation and replication of our study.

Action Taken: We have added a supplementary file named “S1_Table.xlsx” containing the complete stock list.

Stock Code,Stock Name,Industry (L1),Industry (L2)

000063,ZTE Corporation,Communication,Communication Equipment

000547,Aerospace Development,Communication,Communication Equipment

000561,Fenghuo Electronics,Communication,Communication Equipment

000586,Huiyuan Communication,Communication,Communication Equipment

000665,Hubei Broadcasting,Communication,Communication Operations

000701,Xiamen Xinda,Communication,Communication Operations

000725,BOE Technology Group,Communication,Optoelectronic Devices

000748,Great Wall Information,Communication,Computer Applications

000777,Zhonghe Technology,Communication,Communication Equipment

000839,CITIC Guoan,Communication,Communication Operations

000876,New Hope,Communication,Communication Operations

000938,Unisplendour Corporation,Communication,Computer Equipment

000988,Huagong Tech,Communication,Communication Equipment

002049,Unigroup Guoxin,Communication,Integrated Circuits

002089,Sunsea Telecommunications,Communication,Communication Equipment

002093,Guomai Technologies,Communication,Communication Operations

002104,Hengbao Co., Ltd.,Communication,Computer Applications

002115,Sunwave Communications,Communication,Communication Equipment

002151,BDStar Navigation,Communication,Communication Equipment

002161,Yuanwanggu,Communication,Communication Equipment

002194,Wuhan Fangu,Communication,Communication Equipment

002212,Topsec Technologies,Communication,Computer Applications

002231,Allwin Telecommunications,Communication,Communication Equipment

002261,TOWIN Information,Communication,Computer Applications

002281,Accelink Technologies,Communication,Communication Equipment

002296,Splendor Science & Technology,Communication,Computer Applications

002313,Sunsea IoT,Communication,Communication Equipment

002316,Yalian Development,Communication,Computer Applications

002396,Star-net Communication,Communication,Communication Equipment

002417,Shennan Circuits,Communication,Computer Applications

002465,Haige Communications,Communication,Communication Equipment

002467,263 Network,Communication,Communication Operations

002544,P&T Technology,Communication,Communication Equipment

300050,Dingli Communications,Communication,Communication Operations

300051,35 Internet,Communication,Computer Applications

300052,Zhongqingbao,Communication,Internet Services

300074,Huaping Stock,Communication,Computer Applications

300098,Gaoxin Development,Communication,Communication Equipment

300134,Taffull Technology,Communication,Communication Equipment

300136,Sunway Communication,Communication,Communication Equipment

300166,Beyondsoft,Communication,Computer Applications

300168,Wonders Information,Communication,Computer Applications

300177,Zhonghaida,Communication,Communication Equipment

300209,Tianze Information,Communication,Computer Applications

300226,Shanghai Steel Union,Communication,Internet Services

300245,Tianji Technology,Communication,Computer Applications

300250,Chuling Information,Communication,Communication Equipment

300252,Jinxinnuo,Communication,Communication Equipment

300264,Jachung Vision,Communication,Computer Applications

300288,Langma Information,Communication,Computer Applications

300290,Rongke Technology,Communication,Computer Applications

600050,China Unicom,Communication,Communication Operations

600105,Yongding Stock,Communication,Communication Equipment

600130,Bird Corporation,Communication,Communication Equipment

600198,Datang Telecom,Communication,Communication Equipment

600345,Changjiang Communication,Communication,Communication Equipment

600485,Xinwei Group,Communication,Communication Operations

600487,Hengtong Optic-Electric,Communication,Communication Equipment

600498,FiberHome Telecommunications,Communication,Communication Equipment

600522,Zhongtian Technology,Communication,Communication Equipment

600536,China National Software,Communication,Computer Applications

600570,Hundsun Technologies,Communication,Computer Applications

600588,Yonyou Network,Communication,Computer Applications

600637,Oriental Pearl,Communication,Communication Operations

600640,Guomai Culture,Communication,Communication Operations

600776,Eastern Communications,Communication,Communication Equipment

600797,Insigma Technology,Communication,Computer Applications

600804,Dr. Peng Telecom,Communication,Communication Operations

600845,Baosight Software,Communication,Computer Applications

600850,East China Computer,Communication,Computer Applications

600980,Beikang Technology,Communication,Communication Equipment

603000,People.cn,Communication,Internet Services

603160,Goodix Technology,Communication,Integrated Circuits

000001,Ping An Bank,Financial,Bank

000002,Vanke A,Financial,Real Estate

000006,Shenzhen Zhenye A,Financial,Real Estate

000012,CSG A,Financial,Real Estate

000046,Oceanwide Holdings,Financial,Real Estate

000069,OCT Group A,Financial,Real Estate

000402,Financial Street,Financial,Real Estate

000415,Bohai Leasing,Financial,Other Financial

000416,Minsheng Holdings,Financial,Other Financial

000417,Hefei Department Store,Financial,Other Financial

000419,Tongcheng Holdings,Financial,Other Financial

000420,Jilin Chemical Fiber,Financial,Other Financial

000421,Nanjing Public Utility,Financial,Other Financial

000423,Dong-E-E-Jiao,Financial,Other Financial

000425,XCMG Construction Machinery,Financial,Other Financial

000426,Xingye Mining,Financial,Other Financial

000428,Huatian Hotel,Financial,Other Financial

000429,Guangdong Expressway A,Financial,Other Financial

000430,Zhangjiajie Tourism,Financial,Other Financial

000488,Chenming Paper,Financial,Other Financial

000498,Shandong Road & Bridge,Financial,Other Financial

000501,Wushang Group A,Financial,Other Financial

000503,Guoxin Health,Financial,Other Financial

000506,Zhongrun Resources,Financial,Other Financial

000507,Zhuhai Port,Financial,Other Financial

000509,Huasu Holdings,Financial,Other Financial

000510,Xin Jinlu,Financial,Other Financial

000513,Livzon Group,Financial,Other Financial

000514,Yujian Development,Financial,Real Estate

000063,ZTE Corporation,Communication,Communication Equipment

000547,Aerospace Development,Communication,Communication Equipment

000561,Fenghuo Electronics,Communication,Communication Equipment

000586,Huiyuan Communication,Communication,Communication Equipment

000665,Hubei Broadcasting,Communication,Communication Operations

000701,Xiamen Xinda,Communication,Communication Operations

000725,BOE Technology Group,Communication,Optoelectronic Devices

000748,Great Wall Information,Communication,Computer Applications

000777,Zhonghe Technology,Communication,Communication Equipment

000839,CITIC Guoan,Communication,Communication Operations

000876,New Hope,Communication,Communication Operations

000938,Unisplendour Corporation,Communication,Computer Equipment

000988,Huagong Tech,Communication,Communication Equipment

002049,Unigroup Guoxin,Communication,Integrated Circuits

002089,Sunsea Telecommunications,Communication,Communication Equipment

002093,Guomai Technologies,Communication,Communication Operations

002104,Hengbao Co., Ltd.,Communication,Computer Applications

002115,Sunwave Communications,Communication,Communication Equipment

002151,BDStar Navigation,Communication,Communication Equipment

002161,Yuanwanggu,Communication,Communication Equipment

Comment 2: It is recommended to label the meanings of different types in Figure 3 to make it easier for readers to understand the content of the figure.

Response: We sincerely thank the reviewer for this valuable and constructive suggestion. We agree that clarifying the elements of Figure 3 is crucial for helping readers intuitively understand the distribution and comparison of prediction errors across different methods. In response to this comment, we have implemented the following comprehensive revisions to the figure and its corresponding des

---

## [Decision Letter · Decision Letter 1]

22 Oct 2025

A Multi-Factor Dynamic Time Series Measure for Stock Correlation Analysis

PONE-D-25-44140R1

Dear Dr. Ma,

We’re pleased to inform you that your manuscript has been judged scientifically suitable for publication and will be formally accepted for publication once it meets all outstanding technical requirements.

Kind regards,

Najmul Hasan, PhD

Academic Editor

PLOS ONE

Additional Editor Comments (optional):

Reviewers' comments:

Reviewer's Responses to Questions

**Comments to the Author**

1. If the authors have adequately addressed your comments raised in a previous round of review and you feel that this manuscript is now acceptable for publication, you may indicate that here to bypass the “Comments to the Author” section, enter your conflict of interest statement in the “Confidential to Editor” section, and submit your "Accept" recommendation.

Reviewer #1: All comments have been addressed

Reviewer #2: All comments have been addressed

2. Is the manuscript technically sound, and do the data support the conclusions?

Reviewer #1: Yes

Reviewer #2: Yes

3. Has the statistical analysis been performed appropriately and rigorously?

Reviewer #1: Yes

Reviewer #2: Yes

4. Have the authors made all data underlying the findings in their manuscript fully available?

Reviewer #1: Yes

Reviewer #2: Yes

5. Is the manuscript presented in an intelligible fashion and written in standard English?

Reviewer #1: Yes

Reviewer #2: Yes

6. Review Comments to the Author

Reviewer #1: The authors have made significant efforts to address all of my previous comments. I am satisfied with the responses I have received to all of my remarks. They have added a new paragraph to the introduction discussing modern neural network methods. They have also improved the description of hyperparameter selection in the 'Experimental Setup' section. Furthermore, they have included a new paragraph in the 'Conclusions' section that provides a detailed explanation of factor selection and stability analysis.

Overall, the authors have taken all of my comments into account. I am satisfied with the revised manuscript and consider it to be of high quality. Therefore, I recommend accepting the paper for publication.

Reviewer #2: Thank you for the authors' careful response. Your explanations have basically addressed my questions. Personally, I believe the revised paper has a more rigorous logical structure, clearer thinking, and is easier for readers to understand. The revised article basically meets the journal's requirements, and I recommend accepting it for publication.

7. PLOS authors have the option to publish the peer review history of their article (what does this mean?). If published, this will include your full peer review and any attached files.

Reviewer #1: **Yes: **Andrii Semenov

Reviewer #2: **Yes: **Kaitao Li

---

## [Editor Report · Acceptance letter]

PONE-D-25-44140R1

PLOS ONE

Dear Dr. Ma,

I'm pleased to inform you that your manuscript has been deemed suitable for publication in PLOS ONE. Congratulations! Your manuscript is now being handed over to our production team.

Kind regards,

on behalf of

Dr. Najmul Hasan

Academic Editor

PLOS ONE